# POSITION: SCIENCE IS COLLABORATIVE—LLM FOR SCIENCE SHOULD BE TOO

## ABSTRACT

Modern scientific breakthroughs are increasingly driven by collaborative team effort where researchers combine diverse expertise to tackle interdisciplinary challenges. In this position paper, **we argue that LLM for Science should mirror such cooperative dynamics through Multi-Agent Systems (MAS) instead of pursuing a single omniscient model for all scientific problems**. Following the Canonical Workflow Framework for Research (CWFR), we identify how MAS could benefit each canonical research stage: enhanced reliability for knowledge synthesis by cross-validation, increased creativity for hypothesis formulation via diversifying perspectives, improved robustness for experimental execution through parallel execution and fault-tolerant backup agents, and more diverse opinion in result interpretation and evaluation. We further outline key bottlenecks in the current reality of MAS4Science and future work to address these challenges, concluding with several concrete call to actions for reliably scaling MAS in science from passive tools to active research partners.

## 1 INTRODUCTION

> *"Science is a collaborative effort. The combined results of several people working together is often much more effective than an individual scientist working alone."*
>
> —JOHN BARDEEN, NOBEL LAUREATE[1]

The most transformative scientific discoveries often originate from **research collaboration** that brings together diverse expertise to spark genuine breakthroughs, (Wuchty et al., 2007) used 19.9 million papers over 5 decades and 2.1 million patents to show that **team collaboration** increasingly dominate solo authors in the production of exceptionally high-impact research, even where that distinction was once the domain of solo authors across sciences and engineering, arts and humanities. We argue that **LLM for Science should reflect such collaborative dynamic via Multi-Agent Systems (MAS4Science)** rather than pursuing a single universal agent for all scientific disciplines.

Frontier labs (Anthropic, 2025) have already begun exploring the immense potential of multi-agent systems for complex scientific reasoning tasks, which leverage parallel thinking for more **creative** exploration of diverse solution paths and cross-validation among agents for more **reliable** result synthesis, achieving gold-medal level performance on International Mathematical Olympiad (IMO) (DeepMind, 2025). This paradigm has quickly expanded to research-level scientific applications across many disciplines, including chemistry and materials science (Zheng et al., 2023; Jin et al., 2025; Gustin et al., 2026), physics and astronomy (Xu et al., 2025; He et al., 2025), biomedical (Solovev et al., 2024; Song et al., 2025; Zhang et al., 2025c) and social sciences (Haase & Pokutta, 2025).

**Working Definitions.** As this position paper is intended for a broad audience, we make an effort to avoid math-heavy formulation and instead characterize the key distinction of multi-agent system (MAS) from single-agent system (SAS) using 2 complimentary property based on established literature (Guo et al., 2024):

---

[1] John Bardeen was the only person to have received the Nobel Prize for Physics twice in 1956 (for the invention of transistors) and 1972 (for pioneering the theory of superconductivity). This quote comes from his Nobel Banquet Speech in 1972, Stockholm.

First, MAS enables **parallelism**: multiple agents can work together simultaneously in a shared environment (not necessarily on the same task so long as they are contributing to the same goal). We note that while it's possible to query a single *model* (which is passive and only responds to external control flow) concurrently, it's not the case for *agent* (which is active and can have an internal control flow that allows agents to plan&act (Yao et al., 2022) autonomously), concurrent calls to agents would create separate instances/threads (also known as *sub-agents*), which is effectively creating multiple agents with the same backbone model.

Second, MAS follows **collectivism**: MAS as a whole is impacted by multiple agents collectively in various interaction mechanisms: whether through explicit communication such as multi-agent debate/discussion (MAD), through division of labor where different agents perform different sub-tasks of a large mission, or through aggregation such as majority voting and cross-validation on the same task. In this study, we mainly focus on MAS with *test-time interaction* through explicit coordination (often via MAD) akin to human research collaboration, but we do note that such coordination is not a necessary condition that defines MAS (for example, 3 coding agents working on adding 3 new function to the same codebase may or may not coordinate explicitly if there's no synergy or conflicts in their mission scope).

**Comparison to Related Work.** Existing studies have focused on AI scientist (Tie et al., 2026; Gottweis et al., 2025; Ghafarollahi & Buehler, 2024) or MAS (Raza et al., 2025; Guo et al., 2024; Li et al., 2024; Zhu et al., 2025a) separately, where our work aims to focus on the intersection of MAS4Science not only on one domain or research task (Zhuang et al., 2025; Zheng et al., 2025; Sami et al., 2024; Luo et al., 2025), but offer a more holistic view on MAS during the entire life cycle of a scientific project. To that end, we adopt a principled approach grounded in the Canonical Workflow Framework for Research (CWFR) (Betz et al., 2022) by the Research Data Alliance (RDA)[2] that identifies recurring patterns in canonical scientific workflow, which guides our selection of four key stages common to modern research: (1) Knowledge Synthesis (literature review), (2) Hypothesis Formulation (research design), (3) Experimental Execution (the term *experimental* broadly includes wet-lab experiments as well as derivation&proof for theorists and simulation for computational scientists), and (4) Result Interpretation and Evaluation (peer review) as illustrated in Figure 1.

From Section 2 to Section 5, we go through these four CWFR stages to first present **opportunity** of MAS at each stage (marked as **O1 to O8**) and offers our **recommendations** (marked as **R1 to R8**) to address major bottlenecks in the current landscape. We then discuss alternative views (Section 6), and conclude with a call to actions for 2 key stakeholder groups: AI researchers who develop LLMs for Science and scientific researchers that use them (Section 7).

We intentionally avoid using too many domain-specific case study as the field of LLM4Science is vast and highly heterogeneous (such that case study in one scientific context may not transfer to another, even use cases in the same domain can be very diverse). Therefore we intentionally keep the arguments primarily logical (with empirical evidence) to be widely applicable for most scientific scenarios.

**Stance and Contribution.** Before proceeding to the main sections, we make a few clarifications on our stance: First, we recognize that current MAS are far from perfect and candidly point out key challenges in its current state with recommendations for future work. Second, MAS is a universal solution for all tasks. Indeed, one should not blindly use MAS before carefully probing their cost-effectiveness from multiple dimensions including accuracy, speed, robustness and many more. Our core thesis is a forward-looking one that MAS represents a more promising direction for *open-ended scientific discovery* as it incorporates the collaborative foundations for human scientific success (Wuchty et al., 2007). We believe this is a timely position paper that connects and encourages both AI developers and frontline scientists to join efforts towards realizing the full potential of cooperative AI for Science.

## 2 KNOWLEDGE SYNTHESIS

Knowledge synthesis requires systematically gathering information from multiple sources, validating accuracy and relevance, and integrating these findings into coherent summaries for evidence-

---

[2]www.rd-alliance.org

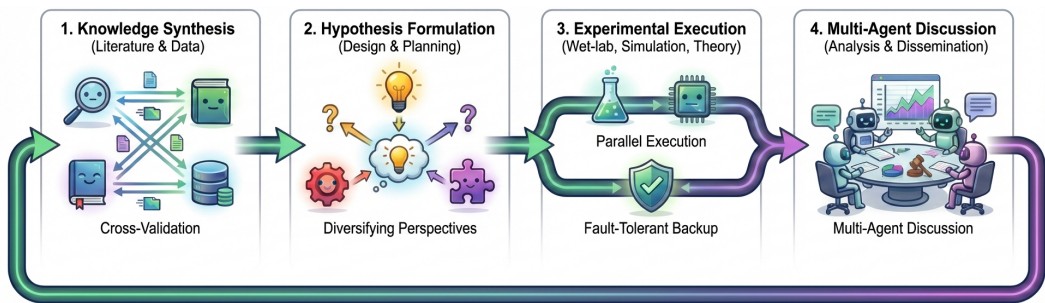

Figure 1: Illustration of the unique advantage brought by multi-agent systems across four CWFR stages in scientific workflows.

based decision-making (Trinh et al., 2025). SAS that consolidate both generation and verification within one model often suffer from hallucination (Kalai et al., 2025; Lin et al., 2025), where the launch of DeepResearch mode in ChatGPT (OpenAI, 2025) (Feb. 2025) coincided with a spike in LLM-hallucinated citations on arXiv (Tramèr, 2025). GPTZero has further reported concerning appearances of such hallucinated citations in venues such as ICLR and NeurIPS (Shmatko et al., 2026). In this section, we first argue how MAS could make knowledge synthesis more reliable by cross-validation and more efficient by parallel processing, and provide concrete recommendations for 2 key failure modes in current MAS for knowledge synthesis, namely error propagation and cross-domain knowledge conflict.

**O1: Cross-validation combats hallucination by independent verification.** SAS often struggle with identifying and rectifying hallucination in its own generation as the same agent who generates errors may not spot them efficiently (Stechly et al., 2024; Valmeekam et al., 2023), also known as *confirmation bias* (Koo et al., 2024; O'Leary, 2025). MAS offers a promising alternative by distributing information aggregation, summarization, and verification across different agents with respective specialized tooling (Trinh et al., 2025): one agent could specialize in wide exploration via internet search for generation, while another specialize in rigorous retrieval, match and verify against credible sources programmatically.

Previous work (Zhu et al., 2024) showcased this approach by using a discriminator agent to help select useful documents from massive noisy data on the internet, steering the generator agent towards more reliable knowledge synthesis in iterative feedback loop. Empirical evidence from biomedical reasoning tasks shows that well-coordinated MAS can outperform SAS when role structure is explicitly defined (Pu et al., 2025; Inoue et al., 2025). AI Urban Scientist (Xia et al., 2025) also demonstrates the power of role separation in MAS by designating specialized agents to access various databases, then validating claims against credible sources. Empirical studies (Shi et al., 2025; Darwish et al., 2025) further substantiates that MAS can significantly reduce hallucinations through collaborative filtering that covers what one agent missed with others' inspection, outperforming self-consistency methods in hallucinations and uncertainty reduction (Feng et al., 2025).

**O2: Parallel processing improves efficiency and independence.** Processing multiple sources in parallel with independent agents accelerates synthesis while reducing anchoring effects inherent in sequential reading, where later evidence is often shaped to fit narratives formed from earlier context.

Indeed, Anthropic has showed that accumulated context can even induce persona drift in LLMs and degrade their "Helpful, Honest and Harmless Assistant" setting (Lu et al., 2026). We further note the nature of current attention mechanism in LLMs means longer contexts often leads to more hallucination from diluted attention (Liu et al., 2023), also known as *context rot* (Hong et al., 2025) as evidenced by many long-context benchmarks reporting unsatisfactory results on frontier models (Bai et al., 2025; Yan et al., 2025b).

MAS could alleviate the pressure of long context by distributing multiple agents for summarizing from distinct sources without impact of context accumulation (Sami et al., 2024), which improves both efficiency and independence of knowledge synthesis using massive literature database. After independent extraction is complete, comparison of these independent summaries could naturally expose inconsistencies and contradictions in literature (where SAS often shape the whole summary

with one consistent narrative (Nogueira et al., 2025)). In scientific inquiry, such conflicts are often informative signals of unresolved research questions for potential breakthroughs, making parallel source processing valuable not only for efficiency but also for identifying gaps in empirical literature to propose new hypothesis.

**R1: Prevent error propagation and snowballing effect.** False information can propagate in MAS with compounding devastating impact (Shen et al., 2025), such snowballing effect is particularly acute when multimodal information such as figures and tables in scientific publications come into play (Yu et al., 2025b). Furthermore, agents usually treat existing publications (especially those in their training data) as authoritative truth by default even though they may also contain errors. Indeed, (Son et al., 2025) have reported that frontier LLMs systematically fail to identify mistakes artificially planted into existing papers and often chose to blindly trust seemingly authoritative sources without cross-validating with other sources on their actual contents. The aggregation of erroneous information from one agent to another is particularly destructive in scientific workflows as they are genuinely needles in a haystack evolving over many rounds of inter-agent communication.

To address error snowballing, MAS architecture should be optimized with selective, not unconditional connectivity and frequent verification at checkpoints: (Shen et al., 2025) demonstrates that optimizing the communication topology by limiting agent connectivity can effectively prevent error propagation while maintaining sufficient information flow for collaborative problem-solving. Beyond architectural improvement, checkpoints could also effectively intercept cascades by tracking information provenance across agent interactions (Zhou et al., 2025) and force iterative cycle of regeneration&verification against primary verifiable sources when agents flag discrepancies in their synthesis (Weng et al., 2025).

**R2: Address cross-domain knowledge conflicts with expertise-weighted discussion.** Cross-domain knowledge synthesis for interdisciplinary research (Aryal et al., 2024; Li et al., 2025b) presents unique challenges, where similar terminologies could bear very different meaning: for example, the term *inflation* has distinct formulation in economics (which describes a general increase in prices and fall in the purchasing value of money) versus cosmology (which describes a phase of exponential space expansion in the early universe). Such poly-semantic concepts could incur two types of failure modes in multi-agent discussions (Khan et al., 2024): If they are trained to attain consensus (Duan & Wang, 2024), they could arbitrarily merge non-compatible concepts across domains simply to reach consensus sooner (which is often used as a proxy reward metric in multi-agent training (Ma et al., 2025)) and subsequently amplify persuasive yet incorrect arguments in an echo chamber (Duan & Wang, 2024). On the other hand, if agents are encouraged to critique other as much as possible, they could deadlock without meaningful summarization.

To address these challenges, future systems should consider incorporating knowledge graphs to explicitly connect cross-domain concepts (Tang et al., 2025b) and assign weights to different agents based on their domain expertise for decision-making (Cherian et al., 2025). This kind of weighted discussion could also better reflect the collaborative dynamic of human collaboration, where domain experts could contribute more in their respective field of expertise. Future MAS could take inspiration from blockchains where such expertise-based weights can be assigned by dynamic reputation tracking via smart contracts, which has been proven to effectively shape collaborative pattern and emergent agent specialization (Qi et al., 2025a).

Last but not least, human scientists with scientific intuition honed by years of domain experience should also play an active role in making judgments at critical junctures to steer MAS towards the right direction in knowledge synthesis (Gaddipati et al., 2025; Spillias et al., 2024), leveraging human-AI collaboration to establish a solid foundation in the first step of scientific discovery.

## 3  HYPOTHESIS FORMULATION

Hypothesis formulation involves the generation of testable explanations for observed phenomena by proposing underlying mechanisms consistent with available evidence. Unlike knowledge synthesis, this stage is inherently speculative and operates in unknown regimes where no objective ground truth exists, which therefore requires a delicate balance between creativity and plausibility.

**O3: Role separation enables more diverse hypotheses.** When single agents generate hypotheses sequentially, each generation could influence the next through inherent dependencies in context and

working memory where promising initial hypotheses bias the agent to make minor amendments on previous ones that narrow down the search space, thereby limiting diversity of proposed hypothesis. (Ke et al., 2025).

On the other hand, MAS maintains role separation through parallel hypothesis exploration where different agents explore different mechanisms simultaneously without seeing others' proposals (Chen et al., 2024b; Wang et al., 2024). This advantage of this mechanism in hypothesis formulation is evident across multiple scientific domains: PriM (Lai & Pu, 2025) employs principle-inspired multi-agent collaboration for material discovery, AstroAgents (Saeedi et al., 2025) generates hypotheses from mass spectrometry data through specialized agent teams. VirSci (Su et al., 2025) further provides empirical ablation study on various team size and rounds in discussion, showing approximately three to five agents and two to three interaction rounds represent a sweet spot to strike a balance between diversity and stability (Ueda et al., 2025). Overall, MAS expands the hypothesis search space by encouraging exploration of distinct causal mechanisms in parallel. This structural diversity increases the likelihood of uncovering non-obvious explanations and reduces premature convergence on a single dominant theory. At the same time, controlled interaction rounds allow hypotheses to be refined without collapsing diversity too early, supporting both creativity and scientific rigor. Controlled experiments indicate that agent diversity, parallelism, and interaction depth exhibit clear sweet spots for ideation quality, providing empirical guidance for setting default configurations (Ueda et al., 2025).

**O4: Multi-agent debate (MAD) enables multi-dimensional examination of hypothesis.** MAS enables structured debate in which agents assume complementary roles and explicitly defend or challenge candidate hypotheses (Bandi & Harrasse, 2024; Du et al., 2024; Khan et al., 2024). For example, one agent advocates a hypothesis with supporting evidence, another probes its assumptions and highlights potential confounders, while a third-party evaluator assesses the relative strength of the competing arguments (Duan & Wang, 2024). This role-based interaction externalizes reasoning that would otherwise remain implicit within a single model and makes the evaluation process more transparent and interpretable. (Du et al., 2024) confirms that MAD improves factuality when different agents bring in genuinely diverse perspectives. Debate combined with code execution has been successfully applied to causal discovery, while also revealing coordination overhead and diminishing returns from excessive deliberation (Le et al., 2024).

Meanwhile, MAD forces each hypothesis to survive systematic counter-arguments, strengthening those that can be consistently defended while revealing internal inconsistencies or unsupported claims in weaker ones, which are subsequently discarded (Yuan et al., 2025). Empirical results further suggest that a small number of critics is sufficient, with three debating agents achieving a favorable balance between argumentative depth and coordination overhead (Ueda et al., 2025). Principle-aware controllers that explicitly balance exploration and exploitation have been shown to substantially improve multi-agent scientific discovery performance (Pu et al., 2025). ARM (Yao et al., 2025) further extends evolutionary discovery to collaboration patterns where the system discovers each agent's reasoning modules through evolutionary search that eliminates modules consistently deferring without improving outcomes.

**R3: Facilitate active participation to converge on actionable hypothesis.** The same independence that enables parallel exploration could also hinder efficient convergence toward a final actionable hypothesis, especially when judging criteria for scientific hypothesis is often subjective in open-ended scientific inquiry without any objective ground truth to lean on. Future MAS need to strike a balance between diversity via sufficiently many threads for exploration and efficiency in having them converge to a few actionable plans via MAD.

We believe the key driver for a successful convergence hinges *active participation* of every agent that represents a diverse hypothesis, otherwise the overall discussion may be skewed with degraded scientific merit. (Note that this is not contradictory with having expertise-weighted discussion as we focus on the fact that each agent should actively engage, but the final results can be a weighted average of everyone's engagement) Researchers have identified emerging *lazy agent* patterns in MAS where some agents dominate the discussion while others merely agree and echo earlier conclusions without materially new contribution (Zhang et al., 2025b), which is likely a downstream consequence of LLM sycophancy (Sharma et al., 2025; Denison et al., 2024) that can evolve into deception even under benign prompts, where more capable models show greater deceptive capabilities (Wu et al., 2025) yet struggle to detect others' lies (Curvo, 2025).

This creates an unbalanced dynamic where the collective effort of MAS could be (too) easily impacted by one influential actor, who may actively deceive to gain more agreements from other agents and dominate the discussion. Potential solvency to this problem includes developing more heterogeneous MAS with different backbone models, contexts and system settings with tool access (Ye et al., 2025) to avoid over-simplistic convergence on similar model/context prior, as well as incorporating merit-based credit mechanism (Zhang et al., 2025b) that encourages active engagement (judged by a third-party audit agent (Duan & Wang, 2024) to assign credit based on their material contribution (improving accuracy of such audit to distinguish genuine contribution from empty arguments remain an open direction for future work).

**R4: Assess quality of formulated hypothesis with uncertainty.** The assessment of hypothesis quality is highly challenging, typically requiring not only objective evidence but also subjective judgment based on *scientific intuition* honed by years of domain experience to see if a hypothesis is worth pursuing from many aspects. While it's still open question as to whether such *scientific intuition* can emerge within AI systems, we mainly argue that MAS can better reflect the multi-dimensional nature of quality assessment in hypothesis formulation.

Specifically, different agents could use different sets of criteria to assess hypothesis quality in various aspects (taking the example of research on a new material: chemical/thermodynamic stability, mechanical strength, manufacturing scalability and environmental impact all need to play a role in hypothesis assessment). Each agent can focus on one concrete aspect without interference at first, and a central evaluator agent can take over at the end for final assessment.

PharmaSwarm (Song et al., 2025) showcased the strength of MAS in hypothesis-driven drug discovery where each agent access dedicated functionality such as genomic analysis, biomedical knowledge graph and binding affinity prediction. A central evaluator agent then ranks the proposals for new drugs by multi-dimensional metrics including biological plausibility, novelty, in silico efficacy, and safety, which accelerate translational research and deliver high-confidence hypotheses more efficiently than traditional pipeline.

Future MAS should also carefully quantify and attribute uncertainty (which is an inherent property of scientific hypothesis) to each agent's proposal using unified, trustworthy uncertainty quantification framework (Yoffe et al., 2025), by carefully inspecting how uncertainty evolves between different agents (Zhao et al., 2025), we could pinpoint vulnerable points where human scientists should then step in to give key guidance in steering the whole system towards creative, yet also practical and trustworthy hypothesis (Tang et al., 2025a; Ghafarollahi & Buehler, 2024).

## 4 EXPERIMENT EXECUTION

Experiment execution involves translating theoretical hypotheses into concrete implementations through dry-lab via computational simulation and wet-lab protocols within physical labs to validate or adjust proposed hypothesis. Scientific progress often requires exploring multiple competing hypotheses in parallel, MAS could facilitate such multi-tasking (Kusne & McDannald, 2023) with independent sub-agents and provide fail-safe redundancy where backup agents can step in when others fail or break, providing better fault-tolerance for the system as a whole to operate normally even when some components malfunction.

It's also worth noting that having *too many tools* often overwhelm SAS as to which one they should use, which subsequently increase misuse and error in tool-using (Lenhard, 2025)). MAS could mitigate such problems by specializing each agent to focus on fewer tools and agent skills (Su et al., 2025).

**O5: Accelerate experimental execution with parallel action.** As we discussed in the definition section, SAS process agentic action requests sequentially that accumulates latency in waiting for task completion and results in substantially longer total execution time. MAS enables parallel action by distributing workload across multiple agents, which significantly reduces overall execution time as the system can exploit concurrent (sometimes asynchronous) actions for independent sub-tasks (Fourney et al., 2024).

Empirical studies (Zhang et al., 2025a) confirms that parallel action in MAS can achieve up to $2.2\times$ speedup while also improving task completion rates on the GAIA benchmark (Mialon et al., 2023).

AgenticSciML (Jiang & Karniadakis, 2025) similarly demonstrates coordinated proposers, critics, engineers, and evaluators operating in evolutionary cycles, where complementary validation steps proceed concurrently to boost overall throughput. Similar multi-agent decomposition appears in AutoLabs (Panapitiya et al., 2025), where different agents handle goal decomposition, stoichiometric computation, and validation in coordinated cycles. By comparison, a single-agent approach must interleave proposing, checking, and revising within one reasoning trajectory, which not only slows execution but also limits opportunities for cross-checking through parallel validation.

**O6: Improve fault-tolerance of MAS by backup agents.** Having multiple agents for the same step provides fault-tolerant alternatives that can substitute for failed agents, ensuring the system continues functioning despite individual failures (Li et al., 2025a). This redundancy allows the system to bypass individual errors tied to a particular reasoning pattern or tool chain and to continue execution through alternative solution strategies. In comparison, SAS typically retries the same failing method, so errors are more prone to block progress as recurring patterns. MAS further provides stronger trouble-shooting because different agents can attempt different tools and methods, increasing overall chance of success. In addition, (Qi et al., 2025a) draws inspiration from blockchain protocols to support transparent agent registration and verifiable task allocation. It further enables dynamic tracking of agent strength through smart contracts, leading to higher task success rates, more stable utility distribution, and emergent agent specialization.

**R5: Prevent concurrent conflicts by proactive coordination.** When agents work simultaneously on shared code without explicit coordination, they can produce incompatible versions that blocks the entire system. For example, one agent may implement functions using NumPy arrays while another relies on Pandas DataFrames, making integration impossible due to mismatched data structures. Conflicts are further amplified when agents modify existing code under incompatible assumptions, such as when one refactors variable names while another simultaneously adds functionality that depends on the original names. The catastrophic consequence of such concurrent conflicts motivates the need for design-time constraints to prevent them preemptively rather than relying solely on post-hoc fixes (Pugachev, 2025).

To address these failures modes for more scalable collaboration, future MAS should implement proactive coordination mechanisms, including exclusive ownership and traceability, merge protocols with mandatory review before integration (Huang et al., 2025), and interface contracts that specify inputs and outputs in advance so agents can work independently on team-level components without jeopardizing system-level dependencies (Tao et al., 2024; Wu, 2025).

**R6: Optimize MAS topology to balance communication overhead with performance gains.** As the number of agents increases in a fully-connected MAS (i.e. any two agents may communicate with each other), each additional agent must coordinate with all existing agents, yielding $O(n^2)$ communication overhead while benefits grow at a much slower pace (Zhang et al., 2024; Yan et al., 2025a). Indeed, previous work has reported diminishing returns as the number of agents exceeds certain context-dependent thresholds (Yang et al., 2025; Kim et al., 2025; junyou li et al., 2024).

To address this challenge, future MAS should reduce communication overhead by designating clear chains of command and ensure only necessary connections are established between worker agents and their respective supervisor, reflecting a clear division of team responsibility (Du et al., 2025) just like human research collaboration. Such hierarchical topologies could reduce coordination complexity from $O(n^2)$ to $O(n \log n)$ or $O(n)$ depending on branching factor as shown in MASTER (Rothfarb et al., 2025), where hierarchical MAS collaboration has shown promise to accelerate material discovery using density functional theory (DFT) workflow.

## 5 RESULT INTERPRETATION AND EVALUATION

Result interpretation involves transforming experimental outcomes into scientific claims by evaluating statistical significance, assessing reliability, contextualizing findings within existing knowledge, and communicating conclusions.

**O7: More diverse input from MAS offers more robust interpretation.** Single agents are commonly trained to optimize self-consistency (Lee et al., 2025; Taubenfeld et al., 2025), which encourages them to maintain one coherent narrative and iteratively adjust it over turns than reconsidering alternative interpretations. On the other side, MAS offers a more robust alternative that designate

different agents to argue for/against certain interpretation, which fosters more engaged discussion that could surface potential weaknesses (Yu et al., 2025a; Zhu et al., 2025b; Jin et al., 2024).

Recent work (Fan et al., 2025; Inoue et al., 2025) further show that both efficiency and accuracy improved with weighted discussion mechanism such as Weighted Iterative Society-of-Experts (WISE), outperforming vanilla MAD across diverse tasks and model configurations. This kind of weighted decision-making could be based on various mechanisms from probabilistic aggregation of annotator error rates (Dawid & Skene, 1979) to peer ranking and consensus-based discussion among diverse models (Li et al., 2023; Chen et al., 2024a), thereby grounding how much each agent's opinion should count towards the final result with their respective strength and weakness.

**O8: Assist peer review with automated checks and more diverse input.** Increasing use of AI for peer review is rapidly becoming an inevitable trend, where frontier venues like AAAI and ICML (Naddaf, 2025) have started pilot trials with AI reviewers (for feedback, not decision). Under this prevalent trend, we argue that SAS review is likely worse as it's one single voice that people can simply copy from (or paraphrase with minimal effort). On the contrary, MAS could enable multiple AI reviewers (Fan et al., 2025) to produce independent reviews (Lan et al., 2024) instead of a single one, which prompts human reviewer/ACs to at least compare and balance various views when making decisions at critical junctures.

Further, multiple agents could also enable efficient checks for several labor-intensive aspects in peer review (most of which are impractical for human reviewers to check by hand due to the increasingly heavy review workload), including flagging potential hallucinations, marking thinly sliced contributions in thousands of submissions and automating reproducibility checks. Indeed, many studies (Siegel et al., 2024; Starace et al., 2025; Kon et al., 2025; Liu et al., 2025b; Ifargan et al., 2024; Seo et al., 2025; Huang et al., 2024) have shown remarkable progress of LLM agents to reproduce experiments from papers. Future MAS bear the potential to significantly accelerate (even automate) the process of reproducibility checks in peer review (which is far too labor-intensive for reviewers to do so manually) and flag any potential issues, thereby motivating better reproducibility practices for the community at large.

We firmly believe that MAS for review should never take over the right to decide, but serve as an information aggregator that helps human reviewers focus on the most important parts of a submission (instead of manually checking every citation or going through hundreds of pages in appendix) to make more informed decisions.

**R7: Prevent metric gaming by mutual oversight.** Reward hacking, formally defined as exploiting the difference between a true reward and a proxy reward (Skalse et al., 2025), often manifests in *metric gaming* where AI competently pursue higher scores in a specific metric by cheating without actually solving the problem. (Bondarenko et al., 2025) reported that frontier reasoning models like OpenAI-o3 and DeepSeek-R1 could actively game the metric by trying to delete opponents rather than winning with genuinely better strategies. This observation further extends to real-world cases in scientific workflows: Sakana AI also reported metric gaming in their AI Scientist (Yamada et al., 2025) system: when experiments took too long and hit timeout limits, AI Scientist simply tried to modify the timeout period instead of optimizing the code, which is a classic example of metric gaming in research environments. Similarly, PostTrainBench (Rank et al., 2025) also revealed that LLMs may attempt to directly train on the test set when tasked with training another model on a given benchmark. Such behavior is particularly destructive in result interpretation, as many experimental results can be interpreted in many ways, some of which can be bended, warped and massaged to arbitrarily fit in certain narrative, leading to misrepresentation of science (Glockner et al., 2024). Future systems should implement scalable oversight (Bowman et al., 2022) that verify each step for genuine problem-solving rather than metric gaming, where stronger model capability can proportionately enhance (instead of breach) oversight.

**R8: Mitigate agentic misalignment by mutual reasoning.** LLM Sycophancy describes the tendency where models tailor responses to input prompts rather than respond objectively (Sharma et al., 2025), leading to bogus interpretation or evaluation contingent on prior context. Empirical evidence have shown that LLMs can progress from sycophancy to full-scale subterfuge (Denison et al., 2024). In MAS, sycophancy could further snowball as agents cater to other agents' input rather than maintaining independent perspectives, exacerbating shared biases and collusion (Motwani et al., 2025). The threat model for collusion encompasses both inadvertent convergence (from shared bias due to

similar training data/paradigm) and adversarial manipulation (from bad actors that mislead collective decision-making (Liu et al., 2025a)).

To address these challenges, mutual reasoning (Qi et al., 2025b) have shown considerable efficacy, where agents actively reason about the states and actions of other agents to enable more transparent coordination. PeerGuard (Fan & Li, 2025) showcased this approach by having each agent evaluates others' response to detect illogical reasoning processes indicative of malicious actors, achieving high accuracy in identifying poisoned agents while minimizing false positives on clean agents. Having peer agents as overseer of each other is an intrinsic advantage of MAS and could inspire future paradigm of more transparent and robust AI control methods (Pecerskis & Smirnovs, 2026).

# 6    CALL TO ACTION

We call on two key group of stakeholders with detailed recommendations: AI researchers that work on developing MAS and researchers that use them in scientific applications.

## 6.1    FOR AI RESEARCHERS DEVELOPING MAS FOR SCIENCE

**Trust but verify.** Build more hallucination-free MAS with provenance-tracked external memory where every output traces back to programmatically verifiable sources (such as CrossRef). Hallucination-free is a *pre-requisite* for trustworthy AI, which many existing DeepResearch tools (OpenAI, 2025) sadly do not satisfy, we especially need rigorous provenance tracking to prevent the catastrophic cascade of hallucination via inter-agent communications. Any intermediate results or actions should be directly attributed to responsible agent(s) in the form of digital signature and reasoning record. Such record must remain transparent for inspection through a user-friendly interface that makes it easy for scientists without AI background to navigate.

**Scale with caution.** Scale carefully from a small number of agents to strike a balance between performance gains vs. growing communication overhead that diminish scaling benefits (Yang et al., 2025; Gao et al., 2025). It is also critical to use proper baselines when calculating such performance gains (for instance, a 3-agent MAS consumes more tokens than pass@3 on a single agent due to communication), we should carefully evaluate SAS vs. MAS with equal compute to see performance gain stem from MAS schema or simply a result of more test-time compute.

**Be careful with over-anthropomorphizing AI.** While human collaboration patterns has already guided previous success like LLM Debate (Khan et al., 2024), we also need to be careful with anthropomorphizing them too much (Deshpande et al., 2023) as AI may interact in ways unseen in human teams, some of which we cannot yet fully understand (Cloud et al., 2025). AI developers should not guide MAS design by simply copy-paste from human organizational theory. Instead, one can always take inspiration yet adopt a rigorous, evidence-based approach that measures quantifiable improvements for each setting. What works for human collaboration may not work for AI, and vice versa.

## 6.2    FOR RESEARCHERS WORKING ON SCIENTIFIC APPLICATIONS

**Clear instruction at each step.** A critical advantage of MAS is that each agent can dedicate to one task (or one aspect of a large task) with user-defined scope and tools. This role separation creates natural compartmentalization for user to clearly track every aspect, which also requires researchers to clearly define the scope, task and toolbox for each agent before putting MAS to work.

**Human judgment at critical junctures.** AI cannot replace human judgment honed through years of domain expertise. Frontline researchers should thoroughly understand both the strengths and caveats of AI tools to use MAS as facilitators rather than substitutes of their own thinking, especially at critical junctures.

**Communicate genuine research needs.** AI developers cannot build truly useful tools without understanding what frontline scientists genuinely need. Current scientific reasoning benchmarks heavily focus on "test-taking" capability using Olympiad questions, yet the ability to solve IMO problems do not necessarily transfer to research activities. We need a inclusive platform to match supply from people who build AI tools and demand of people who use them.

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

# A  APPENDIX

You may include other additional sections here.

