# OpenReview forum: "Position: Science is Collaborative—LLM for Science Should Be Too"
_ICLR.cc/2026/Workshop/FM4Science — ICLR 2026 Workshop FM4Science Poster_

### Official Review · Reviewer_3Y8H · 2026-02-19
**A Broad but Shallow Advocacy for Multi-Agent Systems in Scientific Workflows**

**Rating:** 5
**Confidence:** 3

**Review:**

This position paper argues that LLM-based Multi-Agent Systems (MAS) should mirror the collaborative dynamics of human research teams, organized around four CWFR stages: knowledge synthesis, hypothesis formulation, experimental execution, and result interpretation. For each stage, the authors identify opportunities (O1–O8) and recommendations (R1–R8), concluding with a call to action for AI developers and scientists.

Pros:
Timely topic with a clear, intuitive thesis grounded in real evidence (Wuchty et al., 2007; Anthropic's multi-agent system; DeepMind's IMO work). The breadth of literature coverage (~100+ refs) is impressive for a workshop paper.
Balanced treatment: the paper honestly flags key failure modes—error propagation, lazy agents, sycophancy, metric gaming—rather than being purely advocacy-driven.
Clean structure: the CWFR-based O/R pairing makes arguments easy to follow, and the separated call to action for developers vs. scientists adds practical value.

Cons:
Weak fit with FM4Science scope. The workshop centers on scientific foundation models with physical priors, conservation laws, UQ, and domain-specific architectures. This paper is about generic LLM-based multi-agent orchestration—the arguments apply equally to software engineering or business analytics. There is no serious engagement with what makes scientific MAS distinct (e.g., encoding symmetries, interfacing with simulators/PDE solvers).
No empirical contribution. The paper acknowledges the need for equal-compute SAS vs. MAS comparisons (Section 6.1) but never provides or synthesizes such evidence. Given that Anthropic's own data shows ~15× token overhead for multi-agent systems, the cost-performance tradeoff deserved more than a passing mention.
Missing promised content. The introduction promises an "alternative views" discussion, but no such section exists. For a position paper, seriously engaging with when MAS doesn't help is essential.
Significant typo (p.2, line 97): "MAS is a universal solution for all tasks" almost certainly should read "MAS is not a universal solution," directly contradicting the authors' stated intent.
Surface-level arguments. Each O/R gets roughly one paragraph citing 2–5 papers without deep analysis of why systems succeeded, what failed, or what generalizable principles emerge. The result reads more like an annotated bibliography than a sharply argued position.

Overall: A well-written and comprehensive literature map on MAS for science, but the mismatch with the workshop's focus on domain-specific scientific foundation models, the lack of depth in argumentation, and the absence of new evidence or quantitative analysis place it marginally below the acceptance threshold.

---

### Official Review · Reviewer_mke3 · 2026-02-22
**Good position paper with comprehensive summaries for opportunitis and challenges**

**Rating:** 8
**Confidence:** 4

**Review:**

This position paper argues that LLMs for Science should move beyond single-agent paradigms and adopt multi-agent systems (MAS) to better reflect the inherently collaborative nature of scientific discovery. Structured around the Canonical Workflow Framework for Research, the paper systematically examines the role of MAS across four stages of the scientific lifecycle and provides concrete calls to action for both AI developers and domain scientists.

The paper is well-organized, clearly written, and comprehensive in scope. It does a commendable job synthesizing recent MAS and AI-for-Science literature and framing the discussion at the workflow level rather than focusing on isolated tasks. Importantly, the authors do not uncritically promote MAS; they explicitly discuss risks such as error propagation, metric gaming, sycophancy, and communication overhead, which strengthens the credibility and balance of the argument.

While the paper does not introduce new technical methods or empirical results, this is appropriate for a workshop position paper. Its main contribution lies in providing a structured and timely agenda for MAS4Science and encouraging closer collaboration between AI developers and scientific practitioners.

---

### Meta-Review · Area_Chair_2w2P · 2026-02-27

**Recommendation:** Accept (Oral)
**Confidence:** 5

**Metareview:**

This is a great paper! Strong accept.

---

### Decision · Program_Chairs · 2026-03-03

Accept (Oral)